# Investigation of Heterologously Expressed Glucose-6-Phosphate Dehydrogenase Genes in a Yeast *zwf1* Deletion

**DOI:** 10.3390/microorganisms8040546

**Published:** 2020-04-09

**Authors:** Jürgen J. Heinisch, Johannes Knuesting, Renate Scheibe

**Affiliations:** 1Fachbereich Biologie/Chemie, Universität Osnabrück, AG Genetik, Barbarastr. 11, D-49076 Osnabrück, Germany; 2Fachbereich Biologie/Chemie, Universität Osnabrück, AG Pflanzenphysiologie, Barbarastr. 11, D-49076 Osnabrück, Germany; knuesting@gmail.com

**Keywords:** Baker’s yeast, pentose phosphate pathway, oxidative stress, heterologous expression

## Abstract

Glucose-6-phosphate dehydrogenase (G6PD) is a key enzyme of the oxidative part of the pentose phosphate pathway and serves as the major source of NADPH for metabolic reactions and oxidative stress response in pro- and eukaryotic cells. We here report on a strain of the model yeast *Saccharomyces cerevisiae* which lacks the G6PD-encoding *ZWF1* gene and displays distinct growth retardation on rich and synthetic media, as well as a strongly reduced chronological lifespan. This strain was used as a recipient to introduce plasmid-encoded heterologous *G6PD* genes, synthesized in the yeast codon usage and expressed under the control of the native *PFK2* promotor. Complementation of the hypersensitivity of the *zwf1* mutant towards hydrogen peroxide to different degrees was observed for the genes from humans (*HsG6PD1*), the milk yeast *Kluyveromyces lactis* (*KlZWF1*), the bacteria *Escherichia coli* (*EcZWF1*) and *Leuconostoc mesenteroides* (*LmZWF1*), as well as the genes encoding three different plant G6PD isoforms from *Arabidopsis thaliana* (*AtG6PD1, AtG6PD5, AtG6PD6*). The plastidic AtG6PD1 isoform retained its redox-sensitive activity when produced in the yeast as a cytosolic enzyme, demonstrating the suitability of this host for determination of its physiological properties. Mutations precluding the formation of a disulfide bridge in AtG6PD1 abolished its redox-sensitivity but improved its capacity to complement the yeast *zwf1* deletion. Given the importance of G6PD in human diseases and plant growth, this heterologous expression system offers a broad range of applications.

## 1. Introduction

All cells from bacteria to humans have developed mechanisms to cope with oxidative stress caused by reactive oxygen species (ROS), which are a by-product of respiratory metabolism and cause damage to membranes and DNA [1]. The importance of corresponding signaling pathways that elicit a proper cellular response to ROS accumulation, especially in tumor cells, has been underlined by studies ultimately earning last year’s award of the Nobel prize for physiology and medicine (see [2], and references therein). Glucose-6-phosphate dehydrogenase (G6PD), a key enzyme of the pentose phosphate pathway (PPP) generates NADPH required to detoxify ROS, but also for methionine biosynthesis [3,4]). In human erythrocytes, G6PD is the exclusive source of NADPH and related to the most common human enzymopathic condition [5,6]. G6PD also plays a pivotal role in cancer cell proliferation, cellular signaling, embryogenesis, apoptosis, neurodegeneration [7,8,9], and cardiac dysfunction in mouse models [10]. The human *G6PD* gene resides on the X-chromosome and heterozygosity for some mutations may protect against malaria infections [11]. This correlates with the observation that the malaria parasite, *Plasmodium falciparum*, is very sensitive to oxidative stress, and drugs targeted towards its native G6PD are used to combat infections [12].

The model plant *Arabidopsis thaliana* has six isoforms of G6PD (AtG6PD1-6), of which the first four are localized in the plastids, whereas the isoforms 5 and 6 reside in the cytosol [13]. In a more recent study, it was shown that G6PD4 is in fact encoding a non-functional isoform, belonging to a new group, namely P0, that is activated under stress, forming double heterodimers with G6PD1, and is translocated to peroxisomes [14]. While the cytosolic forms are insensitive to modulations by the cells’ redox state [15], the plastidial isozymes G6PD1-3 undergo light-dark-modulation and are only active in their oxidized state. Such thiol-disulfide exchange reaction allows for the formation of a reversible disulfide bridge between two regulatory cysteine residues located in a unique loop [16,17]. The reduced form occurs primarily in the illuminated chloroplast where photosynthetic electron flow from reduced ferredoxin, mediated by the ferredoxin-thioredoxin reductase reduces thioredoxin (Trx m). In the light phase, it is largely inactive due to its low substrate affinity [18]. This redox-switch functions due to continuous reoxidation in the light, and rapid oxidation when the electron flow stops upon darkening [19]. The affinity of the oxidized enzyme towards glucose 6-phosphate (G6P) is largely increased [20], so that the enzyme is fully active under physiological G6P concentrations [21]. As a consequence, light-inactivation of the plastidial oxidative PPP prevents futile cycling when the Calvin–Benson cycle enzymes of the reductive PPP are activated for assimilate CO_2_ during the day, thus precluding simultaneous glucose oxidation [19]. However, the physiological characterization of a single G6PD isozyme in the native plant is hampered by the overlapping activities of the six isozymes. The reductive power generated by G6PD is of central importance to plants, since it ensures survival under conditions of salt, drought, and other types of oxidative stress [22,23,24]. Accordingly, transgenic plants with increased levels of G6PD in plastids, or isoform replacement by cytosolic expression of the gene encoding the plastidic isoform P2 in tobacco, which is insensitive to feedback-inhibition by NADPH, were more resistant towards oxidative stress [25,26]. Moreover, the increased reducing power required during nitrate assimilation is also met by elevating the activity of the oxidative part of the PPP in plastids [27,28].

In the Baker’s yeast *Saccharomyces cerevisiae,* mutants in the single gene encoding G6PD (*ZWF1* for “Zwischenferment”, derived from early studies of the PPP) were first reported by [29]. The gene was cloned and sequenced in the early 1990s and employed to construct deletion mutants [30,31]. These mutants were hyper-sensitive towards oxidative stress, such as the addition of hydrogen peroxide, diamide and phenylhydrazine, and are auxotrophic for methionine, underlining the predominant role of G6PD in the production of cytosolic NADPH, since besides its role for glutathione reduction to scavenge ROS, three molecules of NADPH are required for the synthesis of one molecule of methionine [4]. The PPP has been suggested to provide the fastest response of *S. cerevisiae* towards oxidative stress, exerted within seconds after its application [32]. This involves a complex interplay between glycolysis and the PPP, which are linked via the production of fructose-6-phosphate and glyceraldehyde3-phosphate by the transaldolase and transketolase reactions within the non-oxidative part of the PPP [33]. Yet, the contribution of the PPP to energy production differs significantly between different yeasts. While under standard growth conditions, less than 5% of the glucose is channeled into the oxidative PPP in *S. cerevisiae* [34], mutant studies in the milk yeast *Kluyveromyces lactis* indicate that the PPP and glycolysis are equally important for its metabolism [35,36].

Oxidative stress triggers a fast inactivation of glyceraldehyde-3-phosphate dehydrogenase (GAPDH) and triosephosphate isomerase (TIM) in the glycolytic pathway, concomitant with activation of NADPH production by Zwf1, as observed both in *S. cerevisiae* and in *Caenorhabditis elegans* [37]. According to an initial hypothesis, the flux through Zwf1 would be due to the accumulation of upstream glycolytic metabolites. However, based on metabolome analyses performed with human cells, it is now believed that the inhibition of glycolysis at the triosephosphate level merely provides a means of recycling of glucose-6-phosphate as a substrate for G6PD [38,39]. In this scenario, inhibition of G6PD by NADPH and ATP, which reduces the efficiency of the G6PD reaction to less than 1% of its maximal capacity in human erythrocytes [40], would be alleviated by the reduced concentration of NADPH as it is channeled towards the reduction of glutathione and the scavenging of ROS [38]. Within this regulatory system, glycolytic GAPDH appears to be most sensitive towards oxidation at its catalytic cysteine residue, thus creating a highly responsive thiol-switch to reroute metabolism as an additional function [41,42].

We here expressed G6PD genes from several organisms in a *zwf1* deletion strain of *S. cerevisiae*. Besides the opportunity to study the biochemical properties of the encoded enzyme in the context of a heterologous expression system, this provides the basis for the screening of genes for special purposes to be used in synthetic biology and biotechnology. Moreover, it will allow high-throughput screens for the discovery of drugs targeting G6PD. Importantly, in the genetic background of the strain used herein, a *zwf1* deletion impairs vegetative growth under standard growth conditions, increases the sensitivity towards oxidative stress, and affects chronological lifespan.

## 2. Materials and Methods

### 2.1. Strains, Media, and Culture Conditions

For construction of deletions in *S. cerevisiae,* the diploid strain DHD5 (*MATa/MATα ura3-52/ura3-52 leu2-3,112/leu2-3,112 his3-11,15/his3-11,15*; [43]) was employed and haploid segregants were obtained from tetrad analyses. The parental haploid strain HD56-5A (*MATα ura3-52 leu2-3,112 his3-11,15*; [44]) served as a wild-type control in different experiments, and is actually one of the parental strains of the commonly employed CEN.PK series [45].

Rich media were based on yeast extract (1% w/v), peptone (2% w/v) from Difco Laboratories Inc., Detroit, MI, USA, with glucose (2% w/v) as a carbon source (YEPD). Synthetic media were prepared with Difco yeast nitrogen base with ammonium sulfate as described in [46], with the addition of amino acids and bases using a mixture provided by MP Biomedicals (Eschwege, Germany; CSM-His-Leu-Trp-Ura) supplemented as required for selection of plasmids or deletion markers, and 2% glucose (w/v) as a carbon source (SCD). A quantity of 200 mg/L of G418 was used for selection of the *kanMX* marker. Hydrogen peroxide was added at the concentrations indicated to induce oxidative stress. For the preparation of solid media, 1.5% agar was added prior to sterilization. Hydrogen peroxide was added to provoke oxidative stress at a concentration of 1.75 mM. Yeasts were incubated at 30 °C, with constant agitation at 180 rpm for liquid cultures. Tetrad analyses were performed on YEPD plates using zymolyase 20T for digestion of the ascus walls and a Singer Instruments micromanipulator as described in [47]. Plates were incubated for 3 days at 30 °C and colony formation was documented by scanning. Brightness and contrast were adjusted for entire plates using Corel Photo Paint, prior to selection of exemplary four tetrads, each. A minimum of 18 tetrads were examined for each cross.

For work with *E. coli*, strain DH5α (Invitrogen, Karlsruhe, Germany) was employed, grown at 37 °C in LB medium (0.5% yeast extract, 1% Bacto peptone, 0.5% sodium chloride), supplemented with ampicillin (50 mg/L) for plasmid selection.

### 2.2. Serial Drop Dilution Assays

To examine phenotypes of yeast strains in serial drop dilution assays, cells were grown overnight in selective medium. The cultures were diluted to an OD_600_ of 0.25 and again grown for 3–5 h. Exponentially growing cells were adjusted to an OD_600_ of 0.1 with fresh medium and dilutions from 10^0^ to 10^−3^ were prepared. A quantity of 3 µL of each dilution was spotted onto plates with the indicated compositions and incubated for 2-3 days at 30 °C. Growth was documented by scanning the plates, adjustment of brightness and contrast of the entire images, and trimming using Corel Photo Paint.

### 2.3. Construction of Deletion Mutants, Cloning, and Tagging of Genes

A *zwf1* deletion in *S. cerevisiae* was obtained by one-step gene replacement [48] by directly introducing a PCR product obtained with the primer pair 16.235/16.236 (Table 1) and pUG6-hh^-^ (a derivative of pUG6 described in [49], in which the two HindIII sites were eliminated each by single nucleotide exchanges) as a template into the diploid strain DHD5, selecting for G418 resistance. The correct substitution of one of the *ZWF1* alleles in the heterozygous diploid was verified by PCR with the flanking oligonucleotides 16.232/16.233 (Table 1), and the respective transformant was sporulated. The haploid segregant HOD269-1C (*MATa ura3-52 leu2-3,112 his3-11,15 zwf1::kanMX*) was obtained from a standard tetrad analysis performed on a YEPD plate [50]. Likewise, a *rpe1* deletion, supposed to be synthetically lethal with a *zwf1* defect, was constructed by PCR-mediated homologous recombination, using pJJH1287 as a template for amplification of the *KlLEU2* marker with the oligonucleotide pair 19.139/19.140 (Table 1). pJJH1287 is a derivative of pUG6 mentioned above, in which the *KlLEU2* open reading frame was placed between the *TEF2* promoter and terminator sequences, substituting the *kanMX* resistance marker. Transformants were selected for growth on synthetic medium lacking leucine, verified by PCR with the flanking oligonucleotides 19.137/19.138 (Table 1) and subjected to tetrad analysis to yield haploid segregants carrying the deletion. In order to assess the biological function of heterologously expressed *ZWF1* homologs, HOD408 (*MATa/MATα ura3-52/ura3-52 leu2-3,112/leu2-3,112 his3-11,15/his3-11,15 RPE1/rpe1::KlLEU2 zwf1::kanMX/ZWF1*) was then constructed by crossing. This diploid strain, which is heterozygous for both the *zwf1* and the *rpe1* deletion, was used as a recipient for transformation with different plasmids carrying *URA3* as a selection marker. Transformants were sporulated and subjected to tetrad analysis. Growth of segregants carrying both deletion markers, *kanMX* and *KlLEU2* together with the plasmid-borne *URA3* gene, was taken as evidence that the respective heterologous gene can complement the *zwf1* defect. In a similar fashion as described for *RPE1*, the yeast *NQM1* gene was substituted by the *KlLEU2* marker by a PCR product obtained from pJJH1287 with the primers 19.143/19.144 in the diploid strain DHD5. Transformants verified with the flanking primers 19.141/19.142 were sporulated and subjected to tetrad analysis to obtain a haploid segregant carrying the *nqm1* deletion for crossing with the *zwf1* deletion.

Plasmids expressing *ZWF1* from *S. cerevisiae* or its homologs from other organisms were obtained by cloning either PCR products (for *S. cerevisiae* and *K. lactis*) or synthetic DNA-fragments under the control of a modified constitutive *PFK2* promoter into the centromeric plasmid pJJH2064 [51]. Plasmids employed with their pJJH numbers are listed in Table 2, and their complete sequences are available upon request. Specifically, to clone the *ZWF1* genes encoding the two yeast enzymes, the one from *S. cerevisiae* was amplified by PCR with the primer pair 16.234/17.036, and the one from the milk yeast *Kluyveromyces lactis* with the primer pair 17.309/18.002 (see Table 1 for oligonucleotide sequences), both using genomic DNA prepaired from the respective strains as template. For cloning of heterologous *G6PD* genes, their coding cDNA sequences were obtained (accession numbers given in Table 2) and synthesized in the yeast codon usage (Thermo Scientific GeneArt synthesis), introducing a unique BamHI site prior to the ATG start and a unique HindIII site following the translation stop codon. Mutations leading to exchanges of specific amino acid residues in the encoded proteins for AtG6PD1 were also introduced by synthesis of DNA fragments with convenient restriction sites for cloning into the original plasmids carrying the respective wild-type genes. All open reading frames were verified by Sanger sequencing (Seqlab, Göttingen, Germany) prior to their introduction into the yeast recipient cells.

For detection in Western blot analyses, 3xHA-tags were introduced by in vivo-recombination into recipient plasmids linearized with HindIII, dephosphorylated with alkaline phosphatase and co-transformed with PCR products obtained from pFA6a-3HA-SkHIS3 [52] as a template for amplification with 17.046 as a primer with homologies to the vector sequences located 3′ to the genes and the appropriate primers generating in-frame fusions to the respective ORFs, as listed in Table 1.

Nucleotide sequences complementary to the template for PCR reactions are given in capital letters, others in small letters. If the latter comprise more than 35 nucleotides, they provide the region for homologous recombination of the PCR product with the target sequences either in the vector (for 3HA tags) or the yeast genome (for construction of deletions). Restriction sites introduced with the primers into the PCR product are underlined. “for” and “rev” in the name, designating the respective forward and reverse primers for each pair. All forward primers used for tagging with 3HA were employed in conjunction with 2064revHA (17.046) to obtain PCR products suitable for recombination with the target vectors listed in Table 2. AtG6PD1for3HA (18.205) was used to tag the wild-type gene as well as the three mutant variants constructed. For cloning of *KlZWF1*, the reverse primer was designed to elimate an internal BamHI site of the gene by silent nucleotide exchanges (indicated by the small letters ct in the run of capital letters). “del5” and “del3” designate the forward and reverse primers for deletion of the respective genes.

### 2.4. Enzyme Assays and Western Blot Analysis

Glucose-6-phosphate dehydrogenase activities were determined routinely from yeast crude extracts prepared by breaking of cells with glass beads and centrifugation, as described previously [53]. Alternatively to the preparation of crude extracts from fresh overnight cultures, we verified that specific G6PD activities did not suffer from freezing the washed harvested cells at −20°C for several days. Glass beads were then added to the frozen pellet and preparation of crude extracts followed the standard protocol from then on. Tris-HCl buffer, pH 7.5, was employed for preparation of crude extracts and enzymatic determinations. G6PD activities were determined by following the kinetics of NADP reduction at a final concentration of 0.4 mM, added from a 40 mM stock solution in extraction buffer, in a DU800 spectrophotometer (Beckman-Coulter, Krefeld, Germany) at 340 nm and 30 °C. Glucose-6-phosphate was added as a substrate at a final concentration of 2 mM, after determination of the extinction difference obtained by crude extract in the absence of the substrate. For treatment with dithiothreitol (DTT), 2 µL of a 100 mM stock solution was added to 200 µL of crude extract and incubated for 20 min at room temperature prior to enzymatic determinations. Likewise, diamide was added to a 200 µL aliquot of crude extract at a final concentration of 0.5 mM for 20 min at room temperature to provide an oxidative environment. All chemicals used for enzyme tests were obtained from Sigma-Aldrich Chemie GmbH, Munich, Germany.

For Western blot analyses, crude extracts prepared as above were mixed with SDS sample buffer (final concentrations: 60 mM Tris/HCl, pH 6.8, 10% glycerol, 2% SDS, 0.005% Bromphenol Blue) and heated to 98 °C for 5 min, prior to separation on precast 10% SDS-polyacrylamide gels (LTF-Labortechnik GmbH and Co KG Wasserburg, Germany). Blotting and preparation for immunological detection was done as explained in [54]: after completion of electrophoresis, gels were transferred to a nitrocellulose membrane (Whatman GmbH, Dassel) with the “Trans-Blot SD Semi-Dry Electrophoretic Transfer Cell” (Bio-Rad). Membranes were blocked for 1 to 3 h at room temperature with TBST (20 mM Tris, 150 mM NaCl, pH adjusted to 7.6 with HCl, 0.05% Tween 20) containing 3% bovine serum albumin (BSA, Roth, Karlsruhe, Germany). 3xHA-labelled G6PD enzymes were detected with an anti-HA mouse antibody (kindly provided by Anja Lorberg, Osnabrück), which was diluted 1:10000 in TBST with 3% BSA. An anti-mouse antibody from goat coupled to IRDye 800CW (Li-Cor Biosciences, Lincoln, NE, USA; diluted 1:5000 in TBST with 3% BSA) was used as a secondary antibody. As an internal loading control, the phosphofructokinase subunits were detected using an anti-PFK polyclonal antiserum from rabbit at a dilution of 1:10.000 [55], and an anti-rabbit IgG antibody from donkey coupled to IRDye 700DX (Rockland Immunochemicals, Gilbertsville, PA, USA, diluted 1:5000 with 3% BSA in TBST) as a secondary antibody. The Odyssey infrared imaging system was used for detection and quantification of signals (LI-COR Biosciences GmbH, Bad Homburg, Germany). For quantification, G6PD signals were determined relative to the control PFK signals. ScZwf1 (G6PD from *S. cerevisiae*) was set at 100% and normalized signals from the heterologous enzymes were calculated as percentages of this value.

### 2.5. Determination of Chronological Life Span

Precultures of cells of strain HD269-1C carrying *CEN/ARS* plasmids with the genes indicated were grown overnight at 30 °C in 2.5 mL of SCD medium lacking uracil for plasmid selection to reach an early stationary phase. They were diluted 200-fold in 25 mL of fresh medium in 100 mL Erlenmeyer flasks and further incubated with agitation (180 rpm/min) for 2–3 days until they reached the stationary phase. Samples were then taken for a total of two weeks at the times indicated, appropriately diluted to yield 50–500 viable cells per 100 µL and plated onto YEPD. The number of viable cells per mL of culture was then calculated from colony counts (colony forming units, CFU) after 3 days of incubation at 30 °C. CFUs determined at day 0 were set at 100% for each culture. CFUs obtained at each time point were then related to this value. Each clone was inoculated in triplicate and mean percentages of survival and standard deviations were calculated from these biological replicates.

## 3. Results

### 3.1. A Yeast System for Heterologous Expression of G6PD Genes

In order to establish a versatile expression system for *G6PD* genes from organisms spanning the kingdoms of bacteria, fungi, plants, and animals, we obtained a deletion mutant in the sole *S. cerevisiae ZWF1* gene, substituting its open reading frame (ORF) for a *kanMX* deletion cassette in one allele of the wild-type diploid strain DHD5 (Figure 1A). To assess the growth phenotypes of haploid segregants carrying the *zwf1* deletion, the diploid was subjected to tetrad analysis and spores were allowed to germinate and grow on rich medium with 2% glucose as a carbon source. In contrast to previous reports, we noted a distinct growth retardation associated with the *zwf1* deletion in this strain background. Thus, a 2:2 segregation for large and small colonies was observed, with the smaller colonies invariably carrying the *zwf1* deletion allele, as confirmed by their resistance towards G418 (Figure 1B). This phenotype was not rescued by the addition of methionine to the rich medium, for which *zwf1* deletions have been reported to be auxotrophic (Figure 1B). One segregant from such an analysis, HOD269-1C, was then chosen for further investigations described below.

### 3.2. Heterologous G6PD Genes from Different Biological Kingdoms Complement a Yeast zwf1 Deletion

The *zwf1* mutant thus constructed was then used as a recipient for heterologous expression of genes encoding G6PD isozymes throughout the biological kingdoms, as listed in Table 2. After the preparation of crude extracts from strains carrying the genes on the *CEN/ARS* vector pJJH2064 under the control of the constitutive *PFK2* promoter, specific G6PD activities were determined and demonstrated the functional expression of genes from the closely related yeast *Kluyveromyces lactis*, the bacteria *Escherichia coli* and *Leuconostoc mesenteroides*, different isozymes of the plant *Arabidopsis thaliana*, and the human *G6PD* gene (Table 2).

The considerable variations in the specific activities of the heterologous enzymes raised the questions of whether i) the genes were properly expressed, and ii) the encoded proteins were stably produced in the yeast *zwf1* deletion. Therefore, all encoded proteins were equipped with a C-terminal 3xHA-tag and subjected to a Western blot analysis, using the heterooctameric yeast glycolytic enzyme phosphofructokinase as a loading control and reference for quantification [55]. It should be noted that specific G6PD activities were not notably affected by the C-terminal tags as compared to those reported in Table 2, whereas the introduction of the same tags at the N-terminus caused a complete loss of enzymatic activity in all cases. As shown in Figure 1C, the C-terminally tagged heterologous enzymes of all but the plant enzymes were produced at approximately half the levels of the native ScZwf1-3HA, with the exception of the one from *L. mesenteroides*, which reached only 20% (protein levels of strains expressing the *A. thaliana* enzymes are discussed below in Section 2.3).

We proceeded by determining the physiological consequences of heterologous *G6PD* gene expression in the yeast *zwf1* deletion. For this purpose, the hypersensitivity of the deletion towards oxidative stress exerted by the addition of hydrogen peroxide to the medium was employed [30,31,56]. Drop dilution assays confirmed that viability of the deletion carrying the expression vector lacking any *G6PD* sequence was strongly impaired in the presence of hydrogen peroxide as compared to the control plate without the stressor or the strain carrying the plasmid with the wild-type *ZWF1* gene (Figure 2). Complementation ability of the heterologous genes largely correlated with the specific activities measured in the crude extracts of the respective strains, with *KlZWF1* and *HsG6PD1* displaying growth like the wild-type, and the *AtG6PD* genes and *EcZWF1* conferring intermediate viability. As an exception, the gene from *L. mesenteroides* appears to lack complementation despite the detectable specific activity in crude extracts.

In order to obtain a more stringent phenotype for the assessment of heterologously expressed genes, we further confirmed that the *zwf1* deletion showed synthetic lethality with an *rpe1* deletion in the genetic background of the strain used herein. Therefore, HOD269-1C (*zwf1::kanMX*) was crossed to an isogenic HD56-5A derivative, in which *RPE1* was substituted by a heterologous marker from *Kluyveromyces lactis* (*rpe1::KlLEU2*), and subjected to tetrad analysis. While segregants of the resulting diploid HOD408 showing resistance to G418 and segregants prototrophic for leucine were readily obtained, double deletions displaying both markers did not produce any viable progeny (Figure 3A). In contrast, a combination of *zwf1* with a *nqm1* deletion, which lacks a paralog of transaldolase of unknown function and was also reported in the *Saccharomyces* genome database (https://www.yeastgenome.org; searched on November 7, 2019) to be synthetically lethal, could not be confirmed, since segregants carrying the double deletions produced viable progeny (Figure 3B). HOD408 was then used as a recipient for plasmids carrying some of the heterologous genes and transformants were sporulated and again subjected to tetrad analyses (Figure 3C). Viability of segregants carrying both deletion markers (*zwf1::kanMX rpe1::KlLEU2*) in conjunction with *URA3* as an indicator of the presence of the respective plasmid confirmed that the wild-type *ZWF1* gene (pJJH2111) from *S. cerevisiae* restored growth to the double deletion, as did the human gene (*HsG6PD1* on plasmid pJJH2223) and *AtG6PD1* (pJJH2125). The latter produced smaller colonies in accordance with its limited capacity to complement the *zwf1* deletion. Neither of the respective segregants grew on medium containing 5-FOA, an antagonist toxic to strains with a wild-type *URA3* gene, confirming that growth depended on the presence of the plasmid. In contrast, no viable segregants of the double deletions were obtained from diploids transformed with pJJH2064 (empty vector) or pJJH2495 (*LcZWF1*).

### 3.3. Mutational Analysis of Plant G6PD1

Results on complementation of the slow-growth phenotype of the yeast *zwf1* mutant and of the specific enzyme activities reported above indicated that G6PD1 from *A. thaliana* lacking its plastidic signal sequence (see legend of Table 2 for details) is barely functional in the heterologous expression system. Since this might be attributed to an inefficient formation of disulfide bridges in the cytoplasm of yeast as opposed to chloroplasts in plant cells, we intended to substitute the two cysteine residues involved by a salt bridge, i.e., the two cysteines were substituted by an aspartate and a lysine residue, respectively (AtG6PD1^C100D/C108K^). Indeed, the respective plasmid was able to rescue the growth defect of the yeast *zwf1* mutant in the presence of hydrogen peroxide much better than the one with the original *AtG6PD1* clone, as demonstrated by drop dilution assays (Figure 4A). However, to our surprise, the increased complementation capacity was also observed for clones in which only the first cysteine residue was substituted for an alanine (AtG6PD1^C100A^), or in a mutant with an alanine–lysine combination (AtG6PD1^C100A/C108K^), not expected to form a salt bridge. This result indicated that it is rather the formation of a disulfide bridge than its absence, which is deleterious for the function of the enzyme in yeast. This cannot be attributed to a stabilization of the mutant proteins in vivo, since they display slightly decreased steady-state levels as compared to the wild-type, as judged from their 3xHA-tagged variants in a Western blot analysis (Figure 4B).

In order to assess the influence of the redox environment on the activity of the AtG6PD1 variants and the other enzymes from *A. thaliana* investigated herein, we determined the specific activities in crude extracts treated with either dithiothreitol (DTT) or diamide as reductive or oxidative conditions, respectively. As expected from previous data on the plant enzyme [17], AtG6PD1 activity depended on its redox state and its activity decreased significantly when incubated with DTT, while those of AtG6PD5 and AtG6PD6 did not (Figure 4C). The yeast G6PD, which was employed as a control, was insensitive to DTT treatment, too. Treatment with diamide generally led to a moderate reduction in specific activities, indicating that the stability of all enzymes in the yeast crude extracts was slightly affected.

### 3.4. Chronological Life Span of S. cerevisiae Is Drastically Reduced by the zwf1 Deletion and Restored by Heterologous Expression of Functional ZWF1 Homologs

In the course of inoculating the *zwf1* deletion strain carrying the vector pJJH2064 without an insertion for several controls, we noticed that they did not survive prolonged periods of storage at 4 °C, in contrast to those carrying the wild-type *ScZWF1* gene. Therefore, the chronological life span of a strain lacking its native glucose-6-phosphate dehydrogenase was investigated and compared to a wild-type strain carrying the vector, and the deletion strain carrying eiher an episomal native *ScZWF1* gene, or the ones encoding the human G6PD1 or the isoform 1 of *A. thaliana*. As evident from Figure 5, the deletion strain lacking G6PD activity rapidly lost viability and formed less than 1% of the colonies counted for the original stationary phase culture within four days of incubation. On the other hand, all other transformants showed a life span comparable to wild-type cells, displaying more than 80% viability after 4 days and taking at least 11 days to decrease to less than 1%. However, it should be noted that the lifespan of the clone carrying the *AtG6PD1* gene appeared to be somewhat reduced as compared to the ones carrying either *ScZWF1* or the human homolog.

## 4. Discussion

In this work, we constructed a deletion mutant of the gene *ZWF1*, encoding the unique glucose-6-phosphate dehydrogenase enzyme (G6PD) in the yeast *Saccharomyces cerevisiae* to assess its suitability as a host for heterologous expression of G6PD homologs from various organisms throughout the biological kingdoms, in order to study their molecular structure and biochemical properties. Unexpectedly, we noted a distinct growth retardation of haploid deletion strains on standard media. This stands in contrast to data on *zwf1* deletions in most other yeast strains, which were reported to grow as wild-type under such conditions [30,31,32,56]. However, the *zwf1* deletion was lethal in another common background of strain Σ1278b [57], underlining the substantial degree of nucleotide polymorphisms observed between commonly employed yeast strains [45]. The intermediate phenotype of the deletion strain used herein thus combines the advantage of being viable with that of reflecting the complementation capacity of heterologously expressed *G6PD* genes directly by the size of the colonies produced upon transformation. A more stringent test for complementation of growth phenotypes is provided by the hypersensitivity of *zwf1* deletions towards oxidative stress, which is shared by our mutant with all viable strains tested in the works cited above. Moreover, we confirmed the synthetic lethality of the *zwf1* deletion with that of the *RPE1* gene [58] in our strain background as yet another means to test for complementation capacity. On the other hand, *zwf1 nqm1* double deletions did not excerbate the growth retardation of single *zwf1* deletions, i.e., they did not display a synthetic defect. This could be explained by the role of the ribulose-5-phosphate-3-epimerase encoded by *RPE1* in the non-oxidative part of the pentose phosphate pathway, thus rendering both parts of the pathway non-functional in the *zwf1 rpe1* double deletion. We, therefore, conclude that the PPP serves essential functions in *S. cerevisiae*, which can be met by either the oxidative or the non-oxidative part through their connections with glycolysis, but not if the two branches are blocked. Accordingly, *NQM1* is supposed to encode a transaldolase of unknown function, which obviously is not required for the non-oxidative PPP, were the *TAL1*-encoded transaldolase works.

Another phenotype observed for the *zwf1* deletion employed in this work is the drastic reduction in chronological lifespan (CLS), which is also alleviated by the expression of heterologous *G6PD* genes from humans and plants. This phenotype is most likely related to the fact that the production of reactive oxygen species (ROS) is a primary cause of ageing from yeast to humans [59], and the G6PD reaction provides the NADPH for its detoxification [3,4]. That CLS has not been assessed for *zwf1* deletions in yeast until now may be attributed to the lack of growth phenotypes under standard conditions discussed above. Our findings thus also provide the basis to study the contribution of G6PD enzymes to ageing in the genetically easily tractable yeast model.

Although all heterologously expressed *G6PD* genes herein complemented the yeast *zwf1* deletion, they did so in various degrees, judging from the two parameters of specific enzyme activity and growth of the transformants in the presence of hydrogen peroxide. These two parameters did not always correlate. For instance, the three isoforms of AtG6PD barely conferred growth to the respective transformants in the presence of oxidative stress, whereas the bacterial enzyme from *L. mesenteroides* did not. Yet, the latter had specific activities that were at least double those of the plant enzymes measured in crude extracts from such transformants. On the other hand, specific activities of G6PD from *E. coli* were in the lower range of those of the plant isozymes, but transformants grew almost like wild-type controls in the presence of hydrogen peroxide. Differences in the growth complementation assays could also not be explained by varying stabilities of the enzymes produced in the yeast recipient strain, since Western blots showed higher steady-state levels for the HA-tagged plant isozymes, which did not complement well, than for the enzyme from the closely related milk yeast *K. lactis*, which did. We conclude that the cytosolic environment within the yeast cell substantially affects the actual in vivo activity of the heterologously produced G6PD chloroplast proteins, as observed in plants [18]. Clearly, the enzymes from human and yeasts work best, followed by the one from *E. coli* and the plant isozymes. The lack of complementation by the other bacterial G6PD, LmZwf1, despite its high specific activity in crude extracts can be attributed to its special cofactor dependence. While all the other enzymes tested herein exclusively use NADP as a cofactor, the enzyme from *L. mesenteroides* is less specific and can also employ NAD [60,61]. Since *S. cerevisiae* does not have a transhydrogenase which could interconvert the reduced forms of these cofactors [62], we assume that redox balance under oxidative stress is severely impaired by the production of this rather unusual heterologous enzyme. The fact, that transformants of the yeast *zwf1* deletion with *LmZWF1* produce even smaller colonies than those with the empty vector indicates that the heterologous enzyme exerts a negative effect already under standard growth conditions.

With regard to the AtG6PD1 isoform of *A. thaliana*, we reasoned that it may not properly form the disulfide bridge in the yeast cytosol, which appears to be crucial in plant chloroplasts for its activity in the dark phase [16]. Although the attempt to substitute the disulfide by an ionic bridge appeared to improve the complementation capacity of the enzyme at first glance, further mutant analyses indicated that the reason for this was rather the lack than the presence of the bridge. As the wild-type AtG6PD enzyme displayed higher steady-state levels in the Western blots of HA-tagged variants, the improved complementation capacity cannot be attributed to enhanced protein stability in the yeast host. Since specific activities in crude extracts were also not drastically changed by the mutations, it must be the interplay of enzyme structure and activity in the yeast cytosol in vivo that causes better performance. Nevertheless, the observation that the activity of the wild-type AtG6PD1 isoform can be influenced by applying reducing conditions to the crude extracts, whilst the mutant enzymes incapable of forming a disulfide bridge do not react, underlines the suitability of the yeast expression system to assess the biochemical properties of single heterologously produced plant isoforms.

Finally, it should be noted that the manipulation of the *ZWF1* gene in *S. cerevisiae* receives increasing attention for several biotechnological applications. Thus, the deletion of the gene improved ethanol production from xylose in a recombinant yeast strain, which was attributed to an increase in NADH production at the expense of NADPH [63]. Vice versa, overexpression of *ZWF1* increased the resistance of a production strain towards furfural, an inhibitor present in the fermentation of lignocellulosic substrates [64]. The reaction is also required for the use of *S. cerevisiae* in fermentations to cope with the ROS produced in the presence of inhibitory phenolic compounds [65]. In addtion, overproduction of *ZWF1* in combination with other genetic manipulations was employed to improve the production of isoprenoids and carotenoids in yeast [66,67], and a *zwf1* deletion carrying the *E. coli* pyruvate dehydrogenase complex was found to accumulate cytosolic acetyl-CoA and thus may serve as an important platform strain for many biosynthetic processes, including the production of biobutanol [68]. In the context of such biotechnological uses, the substitution of the endogenous yeast G6PD by heterologous enzymes with altered biochemical properties, as demonstrated in this work, could well be an approach in metabolic design strategies.

## 5. Conlcusions

We demonstrated that the *zwf1* deletion obtained herein serves as a valuable expression system for heterologous G6PD genes in *S. cerevisiae*, especially when the presence of different isozymes impedes the biochemical analysis in the original host organism. Moreover, the relation of G6PD to oxidative stress response and ageing can be more conveniently studied in the extremely well-established model yeast. We, therefore, believe that this system will have broad applications not only in biotechnology, but also in medicine and agriculture.

## Figures and Tables

**Figure 1 microorganisms-08-00546-f001:**
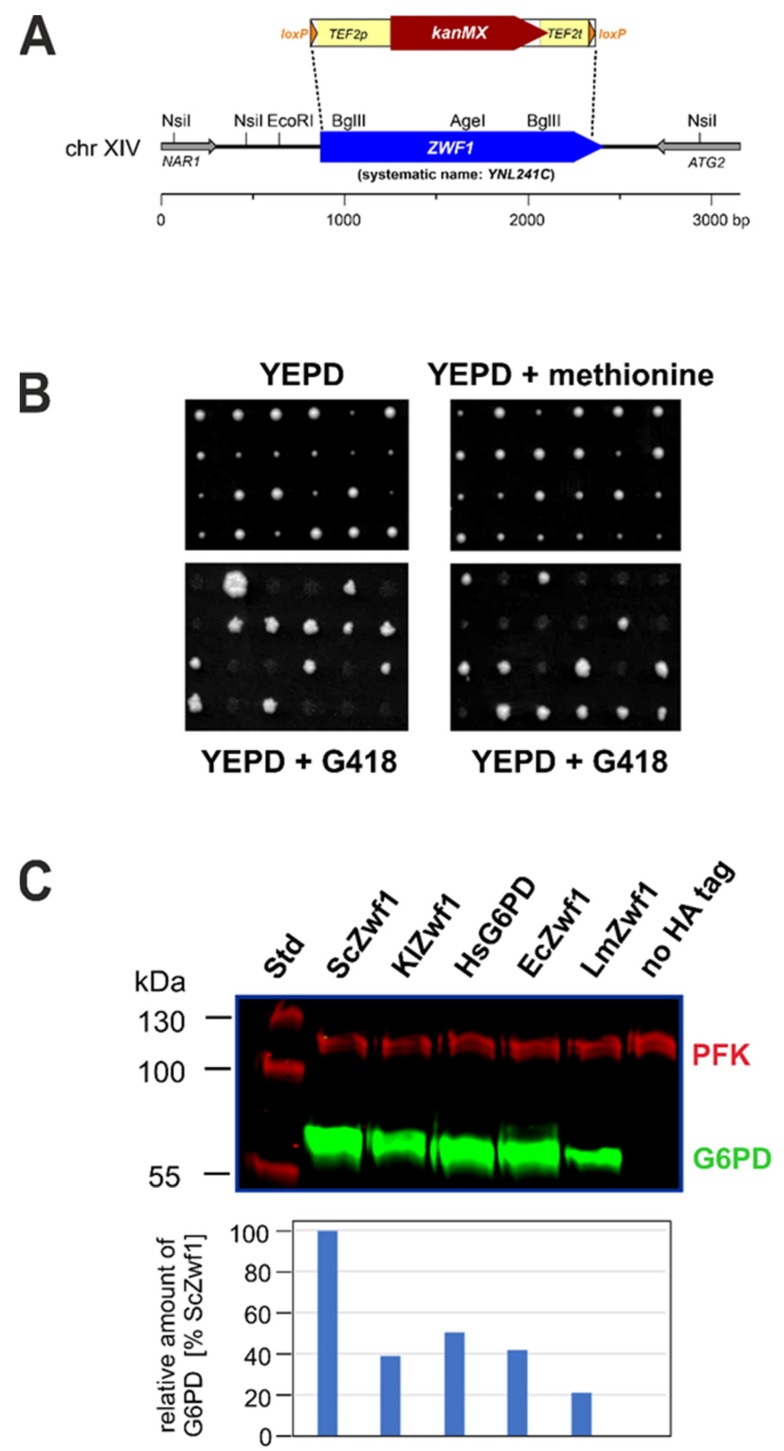
Construction of a yeast *zwf1* deletion strain, growth phenotypes and analysis of heterologously produced G6PD enzymes. (**A**) Schematic respresentation of the wild-type *ZWF1* locus of *Saccharomyces cerevisiae* on chromosome 14 and strategy for its substitution by a *kanMX* marker cassette. Grey arrows respresent flanking genes, the blue arrow the open reading frame of *ZWF1*. Some common restriction sites are shown above. The deletion cassette depicted at the top is flanked by *loxP* sites as targets for the Cre recombinase. (**B**) Tetrad analysis of the heterozygous strain HOD269 (*ZWF1/zwf1::kanMX*) on rich medium (left upper picture) and rich medium supplemented with methionine. The lower panels show growth after replica-plating onto rich medium supplemented with 200 mg/L of G418 (geneticin). Six representative tetrads are shown for each analysis, out of a total of at least 18 tetrads analyzed. (**C**) Western blot analysis of crude extracts from strain HOD269-1C (*zwf1::kanMX*) carrying the genes indicated on a *CEN/ARS* vector. The upper picture shows the results of the blot analyzed with the Odyssey scanner with anti-PFK shown in red and anti-HA detecting the tagged G6PD enzymes in green. Columns in the lower picture show the quantification of G6PD signals normalized to the amount of PFK detected for each lane in the Western blot, setting ScZwf1 at 100%.

**Figure 2 microorganisms-08-00546-f002:**
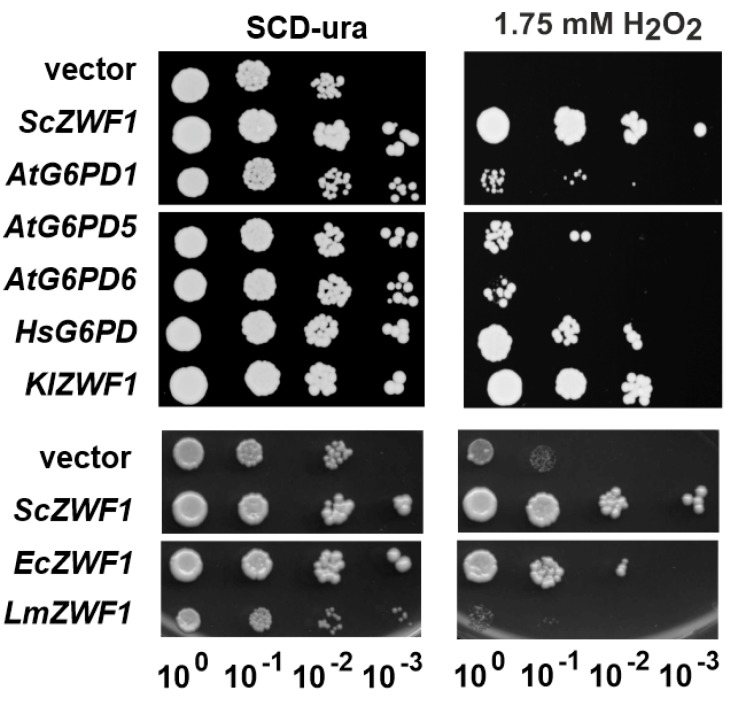
Drop dilution assays of a yeast *zwf1* deletion strain expressing heterologous G6PD genes. Logarithmically growing cultures of strain HOD269-1C (*zwf1::kanMX*) carrying the genes indicated on a *CEN/ARS* vector were diluted in fresh selective medium to an OD600 = 0.1 and subjected to ten-fold serial dilutions as indicated below.A quantity of 3 µL of each dilution was dropped onto the plates indicated and growth was documented after 3 days incubation at 30°C.

**Figure 3 microorganisms-08-00546-f003:**
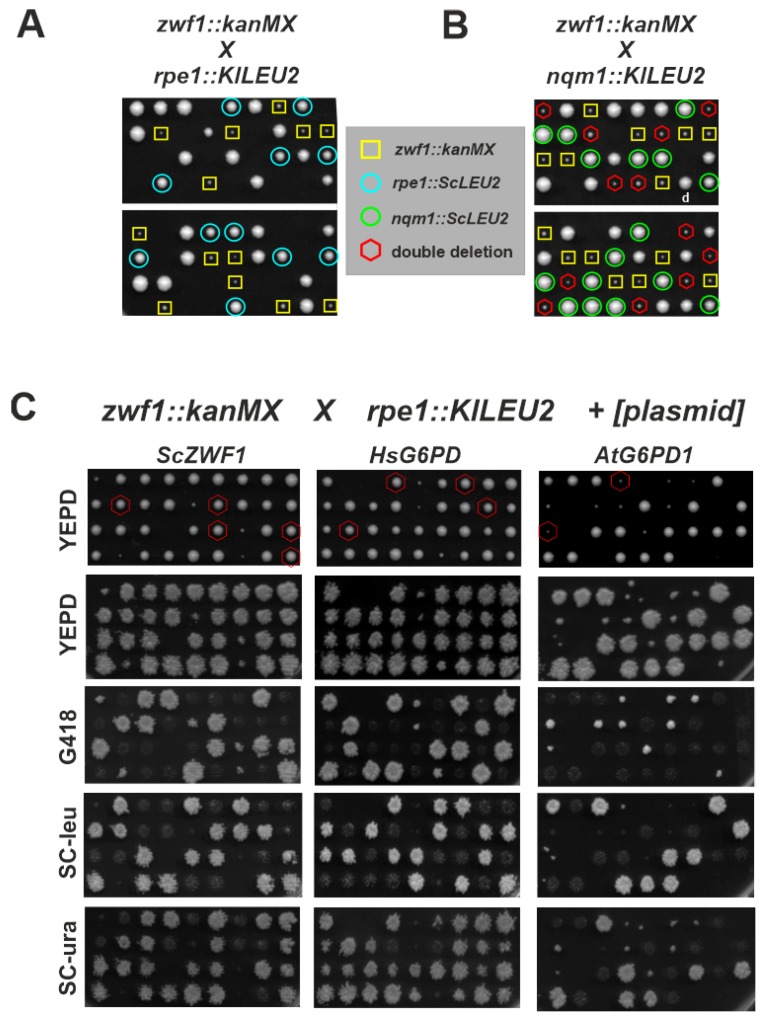
Analysis of synthetic lethality of a yeast *zwf1* deletion and complementation by heterologously expressed genes. Strain HOD269-1C (*zwf1::kanMX*) was crossed to isogenic strains carrying either a *rpe1* deletion (**A**) or a *nqm1* deletion (**B**) and subjected to tetrad analyses on rich medium. Growth was documented after 3 days of incubation at 30 °C. Colored circles and squares highlight single- and double-deletion mutants as indicated. (**C**) The heterozygous diploid strain HOD408 (*ZWF1/zwf1::kanMX RPE1/rpe1::KlLEU2*) was transformed with *CEN/ARS* plasmids carrying the genes indicated, sporulated and subjected to tetrad analyses on rich medium (upper pannels). Growth was documented after 3 days of incubation at 30°C, before replica-plating onto the indicator media shown in the lower pannels. Growth there was documented after 1–2 days of incubation at 30 °C. Red circles indicate viable double deletions complemented by the plasmid-encoded gene.

**Figure 4 microorganisms-08-00546-f004:**
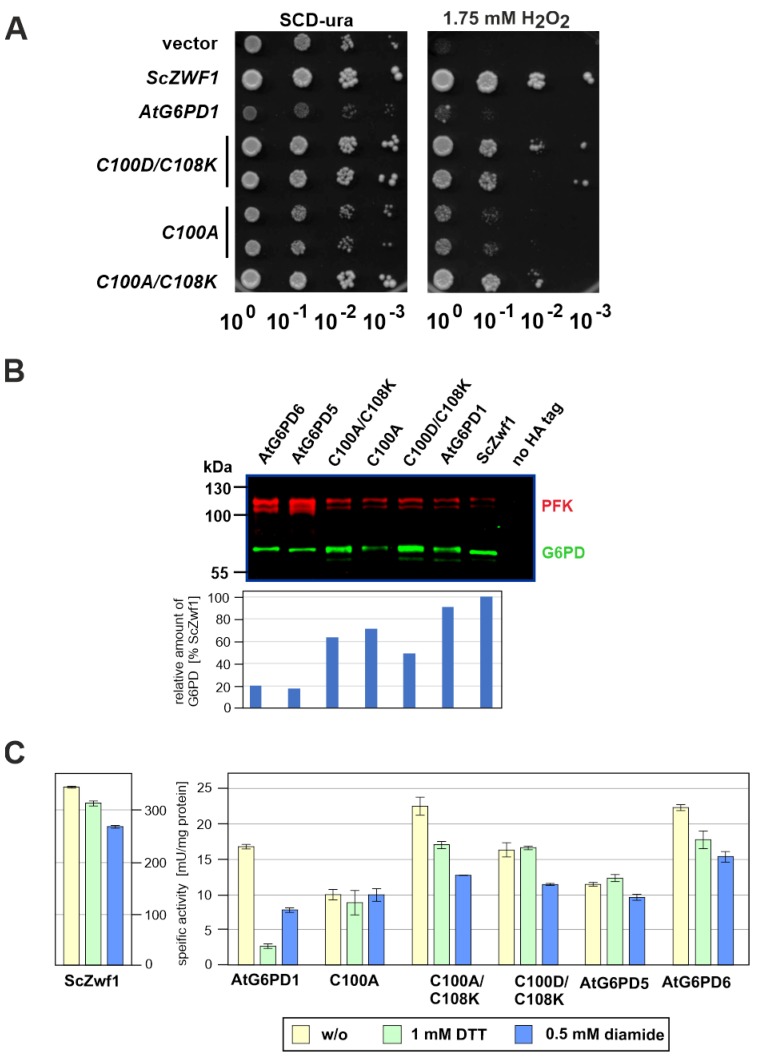
Analysis of a yeast *zwf1* deletion producing different G6PD isoforms from *Arabidopsis thaliana*. (**A**) Logarithmically growing cultures of strain HOD269-1C (*zwf1::kanMX*) carrying either the wild-type *AtG6PD1* gene or the mutated variants indicated on a *CEN/ARS* vector were subjected to drop dilution assays and analyzed as explained in the legend of Figure 2. Strains carrying the empty vector or a plasmid with the native *ScZWF1* gene were employed as controls. (**B**) Western blot of crude extracts from strains producing the indicated AtG6PD isoforms. Antisera and quantification were as explained in the legend of Figure 1C, setting the amount of ScZwf1 to 100%. (**C**) Sensitivity of different G6PD enzymes produced in the yeast *zwf1* deletion strain towards reductive (DTT) and oxidative (diamide) conditions. Specific enzyme activities were either directly determined after preparation of crude extracts (yellow columns) or after pre-treatment of the same crude extracts with 1 mM DTT (green columns) or 0.5 mM diamide (blue columns). Activities were at least determined in triplicate, with the error bars giving the standard deviations of these technical replicates.

**Figure 5 microorganisms-08-00546-f005:**
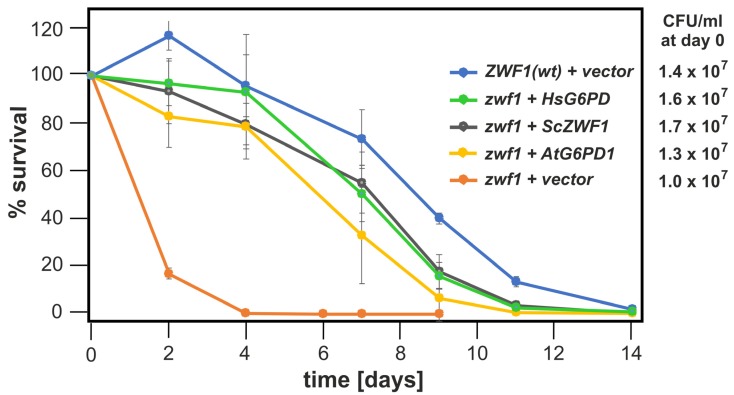
Chronological lifespan of a yeast *zwf1* deletion expressing heterologous *G6PD* genes. The indicated plasmids were introduced into strain HOD269-1C (*zwf1::kanMX*) and cultures were grown to stationary phase in synthetic selective medium. Samples were taken at the time points indicated, appropriately diluted to produce single colonies, plated on rich medium and incubated for three days prior to determination of colony-forming units (CFU). CFUs were set at 100% at day 0, with starting counts indicated at the upper-right corner, and relative viability was determined in three independent cultures, each. Error bars give the standard deviations of % survivors in these three biological replicates.

**Table 1 microorganisms-08-00546-t001:** Oligonucleotides used in this work.

Name (Number)	Sequence (5′ → 3′)
ZWF1atgforBam (16.234)	ggcgggatccATGAGTGAAGGCCCCGTCAAATTCG
ZWF1revHind (17.036)	gcgtgaaagcttGATAAGTACAAGTCCAATCGGACTG
KlZWF1ATGforBam (17.309)	ctacaggatccATGGCTACTCAGTTTGACGAGAAC
KlZWF1revwoBam (18.002)	acacaggaaacagctatgaccatgattacgccaagcttTTACATTTTAGGAGTGGTGACAGGCCATTGGTAGctTCCTGGTTTGGAAAAGG
2064rev3HA (17.046)	acaatttcacacaggaaacagctatgaccatgattacgccaagcttAGGGAGACCGGCAGATCCGCGG
ScZWF1for3HA (18.200)	ttacgcttggcccgtgactaagccagaagatacgaaggataatCGGATCCCCGGGTTAATTAA
KlZWF1for3HA (18.202)	aaaccaggaagctaccaatggcctgtcaccactcctaaaatgCGGATCCCCGGGTTAATTAA
HsG6PD1for3HA (18.205)	ttcaatacgagggtacttacaaatgggttaatccacacaagctgCGGATCCCCGGGTTAATTAA
EcZWF1for3HA (18.201)	gatgattacccgtgatggtcgttcctggaatgagtttgagCGGATCCCCGGGTTAATTAA
AtG6PD1for3HA (18.199)	ttctaagtataacgttagatggggtgacttgggtgaagcaCGGATCCCCGGGTTAATTAA
AtG6PD5for3HA (18.203)	tacatgcaaacccatggttacatttggattccaccaactttgCGGATCCCCGGGTTAATTAA
AtG6PD6for3HA (18.204)	ttacttgcaaacccatggttatatttggattccaccaaccttgCGGATCCCCGGGTTAATTAA
Sczwf1del5 (16.235)	atgagtgaaggccccgtcaaattcgaaaaaaataccgtcatatCTTCGTACGCTGCAGGTCGAC
Sczwf1del3 (16.236)	ctaattatccttcgtatcttctggcttagtcacgggccaaGCATAGGCCACTAGTGGATCTG
ScZWF1forSac (16.232)	gcgtgagctCCTGGTAAGTAAGGTGTAGTTTTG
ScZWF1revSal (16.233)	gtgagtcgacGATAAGTACAAGTCCAATCGGACTG
Scrpe1del5 (19.139)	aagaaggccatttgctaattccaagagcgaggtaaacacacaagaaaaaCTTCGTACGCTGCAGGTCGAC
Scrpe1del3 (19.140)	tatcgtatagtatagagagtataaatataagaaatgccgcatatgtacaaGCATAGGCCACTAGTGGATCTG
ScRPE1forBam (19.137)	CTCGTGgatCCAATAATGAAACTGAAAAGCATG
ScRPE1revHind (19.138)	AAAGAAGcTTCTTTGACTTTGGTTAAGG
Scnqm1del5 (19.143)	cgtaagtcataaaaaataggaaataatcacatatatacaagaaattaaatCTTCGTACGCTGCAGGTCGAC
Scnqm1del3 (19.144)	tggtatatatatatttatatatataagtaggtacctctactcttaatgaGCATAGGCCACTAGTGGATCTG
ScNQM1forXho (19.141)	GCAATCTCGAGAACAATTGCAGGACAGG
ScNQM1revSac (19.142)	gtacggagcTCGGAATTTGATTATACGTCAG

**Table 2 microorganisms-08-00546-t002:** Genes expressed in a yeast *zwf1* deletion and specific G6PD activities.

Strain/Plasmid	Gene	Source Organism	Accession Number ^1^	Specific Activity ^2^ [mU/mg Protein]
HD56-5A/pJJH2064	*ZWF1* (wt)	*Saccharomyces cerevisiae*	CP046094.1	99.33 ± 2.47
			GI:1789112053	
HOD269-1C/pJJH2064	vector	-	-	< 0.5
HOD269-1C/pJJH2111	*ScZWF1*	*Saccharomyces cerevisiae* (strain CEN.PK113-7D)	CP046094.1	312.92 ± 26.60
			GI:1789112053	
HOD269-1C/pJJH2292	*KlZWF1*	*Kluyveromyces lactis*	NC_006040.1	134.43 ± 14.67
			GI: 50313009	
HOD269-1C/pJJH2223	*HsG6PD1*	*Homo sapiens*	BC000337.2	89.03 ± 5.86
			GI: 33991065	
HOD269-1C/pJJH2125	*AtG6PD1* ^3^	*Arabidopsis thaliana*	NM_122970.6 GI:1063734559	12.78 ± 1.43
HOD269-1C/pJJH2224	*AtG6PD5*	*Arabidopsis thaliana*	NM_113644.5	19.38 ± 4.88
			GI: 1063714071	
HOD269-1C/pJJH2249	*AtG6PD6*	*Arabidopsis thaliana*	NM_113644.5	24.38 ± 4.96
			GI: 1063714071	
HOD269-1C/pJJH2494	*EcZWF1*	*Escherichia coli* K12	CP047127.1	20.28 ± 1.05
			GI: 1789840096	
HOD269-1C/pJJH2495	*LmZWF1*	*Leuconostoc mesenteroides*	M64446.1	52.01 ± 1.07

^1^ Nucleotide sequences for *ScZWF1*, *KlZWF1* and *EcZWF1* were extracted from the genome sequences with the given accession numbers from Genbank. Other accession numbers refer to mRNA sequences. All sequences were codon-optimized and synthesized for expression in *S. cerevisiae*, without changes in the primary amino acid sequences of the encoded proteins. ^2^ Specific activities were obtained from at least three independent cultures each, with at at least three technical replicates of enzyme measurements for each crude extract. ± designate the standard deviations. ^3^ For expression in *S. cerevisiae* the *AtG6PD1* gene was synthesized omitting the first 50 codons for the plastidic signal sequence of the protein, starting after an initiating methionine with the amino acid residues FFAEKHSQ.

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
