# Peer review of "Investigation of Heterologously Expressed Glucose-6-Phosphate Dehydrogenase Genes in a Yeast *zwf1* Deletion"

_microorganisms, 2020, doi:10.3390/microorganisms8040546_

Round 1

Reviewer 1 Report

see attached file

Author Response

Fig 3 - Did the authors intend to put two YEPD plates in the image? If so, its not clear in the text, was that the continued growth of the yeasts after transfer onto the other plates? Also can't see any red circles on that image as indicated in the text.

The left image is from a normal YEPD plate, the right image is YEPD with additional methionine. The reason for this experiment was to show that methionine is not limiting in the rich YEPD medium, which could also be a cause for the reduced growth of the colonies carrying the zwf1 deletion.

The authors mention the formation of a disulphide bridge is deleterious to enzyme, this is pretty unusual as its normally lack of disulphide leading to secondary structure collapse which ablates function. Do the authors have any thoughts on this or are their examples of this phenomenom with other antioxidant enzymes?

This is a perfectly valid question and we were also puzzled by this result. However, except for some special conditions in the yeast cytosolic environment we mentioned, which admitedly is a very “diffuse” argument, we do not have an explanation or knowledge of any precedence.

Loss of function of AtG6PD1 with DTT usually suggests a loss of higher molecular weight forms as DTT breaks disulphide bridges. Did the authors run any non-reducing SDS PAGE gels to detect presence of multimeric forms for this or any of the enzymes?

These data have been provided in older works on the plant enzymes in the works cited in the text. In fact, the redox-regulated chloroplast G6PH is unique in these properties, where the reduction of the regulatory intra-subunit disulphide bridge leads to a structural change causing a decreased affinity for the substrate G6P so that this reduced enzyme will be inactive in situ (Scheibe et al., 1989; von Schaewen et al., 1995; Wenderoth et al., 1997; Neé et al., 2014). All other light/dark-modulated chloroplast enzymes are post-translationally inactivated by oxidation meaning that they are inactive in the darkened chloroplast and cycling between reduced and oxidized form allowing for the adjustment of the actual activities according to the demand (Knuesting & Scheibe, 2018). This regulatory principle of reductive inactivation of the OPP cycle enzyme G6PDH in chloroplasts also avoids futile cycling with the reductive pentose phosphate cycle (Calvin-Benson cycle) occuring in the illuminated chloroplast.

Any thought on the over 100% survival in Fig 5? Also could you change the colour of either of the strains currently using a shade of blue, just for clarity. Also any stats to determine if there is a significant difference between AtG6PDi and the others?

Cells were grown from precultures for 2 days to reach stationary phase as explained in M&M. This value was set at 100% for the viable colony counts. The wild-type strain grows extremely well, which may lead to some cell lysis providing nutrients to the remaining cells for some further growth, not observed in the other cultures. This problem cannot be avoided by extended pre-incubation, since the deletion mutant dies rapidly and would therefore start at much lower viable cell counts. Nevertheless, we believe that this is a minor problem, since it only occurs at the second data point after 2 days and only in this one culture.

The light blue shading has now been substituted for green.

Given the error bars, we hesitate to over-interprete the minor differences of AtG6PD1 to the wild-type control.

Reviewer 2 Report

An excellent and enjoyable paper to read, well written, presented well and the data seems to support the conclusions.

A few very minor things

Fig 3 - Did the authors intend to put two YEPD plates in the image? If so, its not clear in the text, was that the continued growth of the yeasts after transfer onto the other plates? Also can't see any red circles on that image as indicated in the text.

The authors mention the formation of a disulphide bridge is deleterious to enzyme, this is pretty unusual as its normally lack of disulphide leading to secondary structure collapse which ablates function. Do the authors have any thoughts on this or are their examples of this phenomenom with other antioxidant enzymes?

Loss of function of AtG6PD1 with DTT usually suggests a loss of higher molecular weight forms as DTT breaks disulphide bridges. Did the authors run any non-reducing SDS PAGE gels to detect presence of multimeric forms for this or any of the enzymes?

Any thought on the over 100% survival in Fig 5? Also could you change the colour of either of the strains currently using a shade of blue, just for clarity. Also any stats to determine if there is a significant difference between AtG6PDi and the others?

Did the authors look at the redox status (glutathione/gssg etc) for the yeast strains expressing different enzymes? Glutathione can function as a thiol donor on occasion. 

Are the authors going to put the constitutively expressed transformants through fermentations with xylose? It would be interesting to see if any of the systems overcome the yeasts inherent co-factor imbalance in the pathway and tendency to accumulate xylitol.

Very minor

Its mL not ml and the same for ul

Author Response

The title should be shortened and changed in some way.

As is stands, it is quite informative, but it looks driving the Reader out-­‐of-­‐ track; more than a methodological paper (as it may be intended by the title), the whole work provide interesting insights in G6PDH regulation. 

We appreciate the suggestion and have rephrased the title to put more emphasis on the investigation of the enzymes, rather than on the methodology.

I must note that the mutational analysis of A.thaliana chloroplastic enzyme is quite brilliant and

could deserve further attention.

P2L36-­‐56. This description of G6PDH could be shortened by one third. I would underline that this part may disorientate those Readers attracted by the current Title of the manuscript (see previous comment); effectively, the “true” intro for a manuscript described by the title starts at L78...

We have shortened the first paragraph of the introduction significantly (now lines 35-46), although with the change of the title it may fit better, as mentioned by the referee. We strongly feel that some of the facts and references should remain, to emphasize the broad biological importance.

P2L57-­‐58: this is not fully correct, G6PD4 encodes for a non-­‐functional isoform P0 activating under stress and directing double heterodimers (2xP1-­‐P0) in peroxisomes as shown before (Meyer et al Plant Journal, 2011).

We have included the data on the P0 isozyme and cited the work mentioned in line 51.

As the Authors correctly noted (P5L157), the N-­‐term of G6PDH is critical in the determination of enzymatic activity. Thus, they should better declare that the chloroplastic A.thaliana isoform AtG6PDH1 studied here was overexpressed devoid of the targeting sequence (I am absolutely confident They did so, otherwise no activity could be have detected!), and possibly the how long was the tag cut away.

The truncation of the signal sequence was already mentioned in the legend of Table 1, although somewhat hidden. We have added the exact start of the primary sequence expressed in yeast there (line 143), as well as a small sentence in the text (line 193), making sure that the readers do not overlook this detail.

Figure 1c should be enlarged if it is possible.

Done. Figure 1C has been enlarged at the expense of Figure 1A and B, which have been reduced a little.

In Figure 5, an in-­‐graph table with the growth starting point values (=100%) for each strain could

be helpful (serving as legend as well).

Done. Besides changing the colour of one curve as requested by reviewer 1, we have added the starting cell counts in the upper right corner.